# The Effect of Removable Orthodontic Appliances on Oral Microbiota: A Systematic Review

**Alessandra Lucchese** [1,2,3,*]**, Chiara Bonini** [4,5]**, Maddalena Noviello** [4,5]**, Maria Teresa Lupo Stanghellini** [6]**, Raffaella Greco** [6]**, Jacopo Peccatori** [4]**, Antonella Biella** [4]**, Elena Tassi** [4]**, Valeria Beretta** [4]**, Fabio Ciceri** [6]**, Paolo Asperio** [7]**, Marta Marcolina** [1,2,3]**, Enrico Gherlone** [1,2,3] **and Maurizio Manuelli** [1,2,3,8]

1. Unit of Orthodontics, Division of Dentistry IRCCS San Raffaele Scientific Institute, 20132 Milan, Italy; marcolinamarta@libero.it (M.M.); alessandrororoccocostantino@gmail.com (E.G.); maurizio.manuelli@gmail.com (M.M.)
2. Unit of Orthodontics, School of Dentistry, Vita-Salute San Raffaele University, 20132 Milan, Italy
3. Unit of Dentistry, Research Center for Oral Pathology and Implantology, IRCCS San Raffaele Scientific Institute, 20132 Milan, Italy
4. Experimental Hematology Unit, Division of Immunology, Transplantation and Infectious Diseases, IRCCS San Raffaele Scientific Institute, 20132 Milano, Italy; piera.cassineri@gmail.com (C.B.); marialuisamerighi@gmail.com (M.N.); luigitorsello@libero.it (J.P.); palinuro09@hotmail.it (A.B.); francesco.mancin@me.com (E.T.); alessandrocosta1995@libero.it (V.B.)
5. Experimental Hematology Unit, Vita-Salute San Raffaele University, 20132 Milan, Italy
6. Department of Medicine, Division of Immunology, Transplantation and Infectious Diseases, IRCCS, 20132 Milano, Italy; chiarabrazzelli@me.com (M.T.L.S.); chiara.garlaschi2709@gmail.com (R.G.); albertosmaria@virgilio.it (F.C.)
7. Division of Maxillo Facial Surgery, Cardinal Massaia Hospital, 14100 Asti, Italy
8. Private Practice Milano, Pavia and Bologna, 20121 Milan, Italy
* Correspondence: lucchese.orthopassion@gmail.com or lucchese.alessandra@hsr.it; Tel.: +39-33-8253-3113

**Abstract:** Background (1): Removable orthodontic appliances may favor plaque accumulation and oral microbe colonization. This might be associated with intraoral adverse effects on enamel or periodontal tissues. The proposed systematic review was carried out to evaluate qualitatively and quantitatively the microbiological changes occurring during orthodontic therapy with removable orthodontic appliances. Methods (2): PubMed, Cochrane Library, Embase, Web of Science, Scopus, Ovid Medline, and Dentistry and Oral Sciences Source were searched. The research included every article published up to January 2020. The Preferred Reporting Items for Reporting Systematic reviews and Meta Analyses (PRISMA) protocol and the "Swedish Council on Technology Assessment in Health Care Criteria for Grading Assessed Studies" (SBU) method were adopted to conduct this systematic review. Results (3): The current study has a moderate evidence, demonstrating that removable appliances do influence the oral microbiota. Significant alterations occur just 15 days after the beginning of therapy, independently from the type of appliance. Furthermore, the levels of oral pathogens decrease significantly or even returned to pre-treatment levels several months later the therapy end. Conclusions (4): This review suggests that orthodontic treatment with removable appliances induces changes to oral microflora, but these alterations might not be permanent.

Protocol: PROSPERO database registration number CRD42019121762.

**Keywords:** oral microbiology; removable orthodontic appliances; oral microflora changes; caries bacteria

## 1. Introduction

Physiologically the human's oral microflora consists of a mixture of organisms, which are common also to other anatomical districts. This bacterial charge is extremely complex, being composed of over 700 different species of bacteria [1–5]. Humans are not randomly colonized and the diverse community that makes up the oral microbiome is finely tuned by nature to protect against diseases, and it is of great importance to maintain its natural diversity. This particular composition depends on numerous factors, some non-modifiable such as genetics, age, sex, change of dentition [6], and some modifiable, including stress, nutrition, dental treatment, and diet [7–10]. The placement of removable orthodontic appliances creates a favorable environment for the accumulation of microbiota components and food residues, which, in time, may cause caries or exacerbate any pre-existing periodontal disease [11–14]. The appliances, both fixed and removable ones [15,16], may interfere with oral hygiene practice and cover considerable parts of the tooth surfaces, so an increase of the total microbial population as well as an altered microflora have been reported in relation to orthodontic treatment [17].

Once dysbiosis occurs, the goal of treatment should be to restore the lost harmonic balance by maintaining good oral hygiene and modifying lifestyle factors such as diet and smoking. The indiscriminate use of antibiotics for the treatment of oral diseases should be avoided to safeguard the beneficial oral microbiota and avoid resistance to antibiotics. Prevention of caries should rely on the use of topical fluoride, and on measures to promote the elimination of the acidic environment, through reduced use of sucrose and acidic drinks (including the sugar-free ones), integration with agents that can reduce the production of acid and/or promote the generation of alkali in dental plaque. For periodontal diseases, therapeutic strategies should aim to mechanically reduce accumulated biofilm by mechanically removing plaque to levels compatible with oral health. This would reduce the inflammation and flow of Gingival Crevicular Fluid (GCF) and promote a favorable microenvironment to support the formation of a balanced microbiome. The role of the oral microbiome is important to prevent oral diseases.

Patients need to be aware of the implications for their oral health when undergoing recommended orthodontic treatment. On the other hand, when a patient accepts to undergo orthodontic treatment, including those using removable orthodontic devices, he should be reminded that it entails a commitment to a higher regimen of attention towards oral hygiene and health in patient's home care [18,19].

The purpose of this review is to investigate the available evidence regarding the association between removable orthodontic appliances and both qualitative and quantitative changes of oral microbiota. Thus, the clinical research questions proposed are:

Do removable orthodontic appliances influence the quality and quantity of oral microbiota? Which are the effects of removable orthodontic appliances on the different bacterial species in the oral cavity?

## 2. Materials and Methods

### 2.1. Protocol

The present study was conducted by the Department of Dentistry at Vita-Salute San Raffaele University of Milan in association with the Unit of Hematology and Bone Marrow Transplantation at San Raffaele Hospital, Milan, Italy. This systematic review was performed in accordance with the guidelines of the Preferred Reporting Items for Systematic Reviews and Meta-Analyses (PRISMA) statement [20,21]. The analysis' methods and inclusion criteria were specified in advance. No funding was given for the realization of the present review. This systematic review followed the PROSPERO protocol and it is registered on its database with the following registration number: CRD42019121762.

### 2.2. Search Strategy

The following electronic databases were searched from their respective sources: PubMed, Cochrane Library, Embase, Web of Science, Scopus, Ovid Medline and Dentistry & Oral Sciences Source; Gray literature was investigated on OpenGray (www.opengrey.eu) and a manual research was conducted on the library of Vita Salute San Raffaele. To create an appropriate research question and review of the literature the PICOS strategy was used: orthodontic patients (patients—P), removable appliances (intervention—I), without orthodontic appliances (comparison—C); oral microbiota (outcome—O) [22,23].

The key words and combinations used in searching the databases were "(Functional appliance OR removable orthodontic appliance OR Frankel appliance OR Bionator OR LM activator OR Twin Block) AND oral microbiology".

Articles published up to January 2020 were included without language and initial date restriction.

### 2.3. Eligibility Criteria

Initially, all articles were selected by title and abstract. Articles present in different databases were considered only once. In a second moment inclusion and exclusion criteria were applied.

The inclusion criteria were: The microbial analysis had to focus on the quality and quantity of changes in the mouth and not on the appliance and the statistical analysis of the studies had to be adequate [24,25]. All the articles included should have a statistical analysis of the results, at least two time points for analysis (with at least one before the beginning of treatment), and at least 10 patients analyzed. Only the studies which analyzed functional removable orthodontic appliances were included, in this way space maintainers, aligners or removable retainers were not considered [26–33].

The exclusion criteria were:

Patients with systemic diseases or under any condition that could influence oral microbiota or periodontal support tissues. Antibiotic therapy within three months before or during the study. No standardization and training in oral hygiene. Studies that did not specify the time of collection of samples [34–36]. Case reports, case series, reviews, and author opinions.

### 2.4. Sudy Selection

To minimize bias, two review authors, with experience in Oral Microbiology (LA) and Functional Orthodontics (MM), analyzed each selected paper and extracted data independently. If data were not clear enough, an attempt was made to contact the author by email. Any disagreement between the two reviewers was resolved by discussion or consultation with a third experienced author (MMa).

The selection of articles was processed according with the PRISMA guidelines (Figure 1 ).

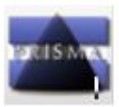

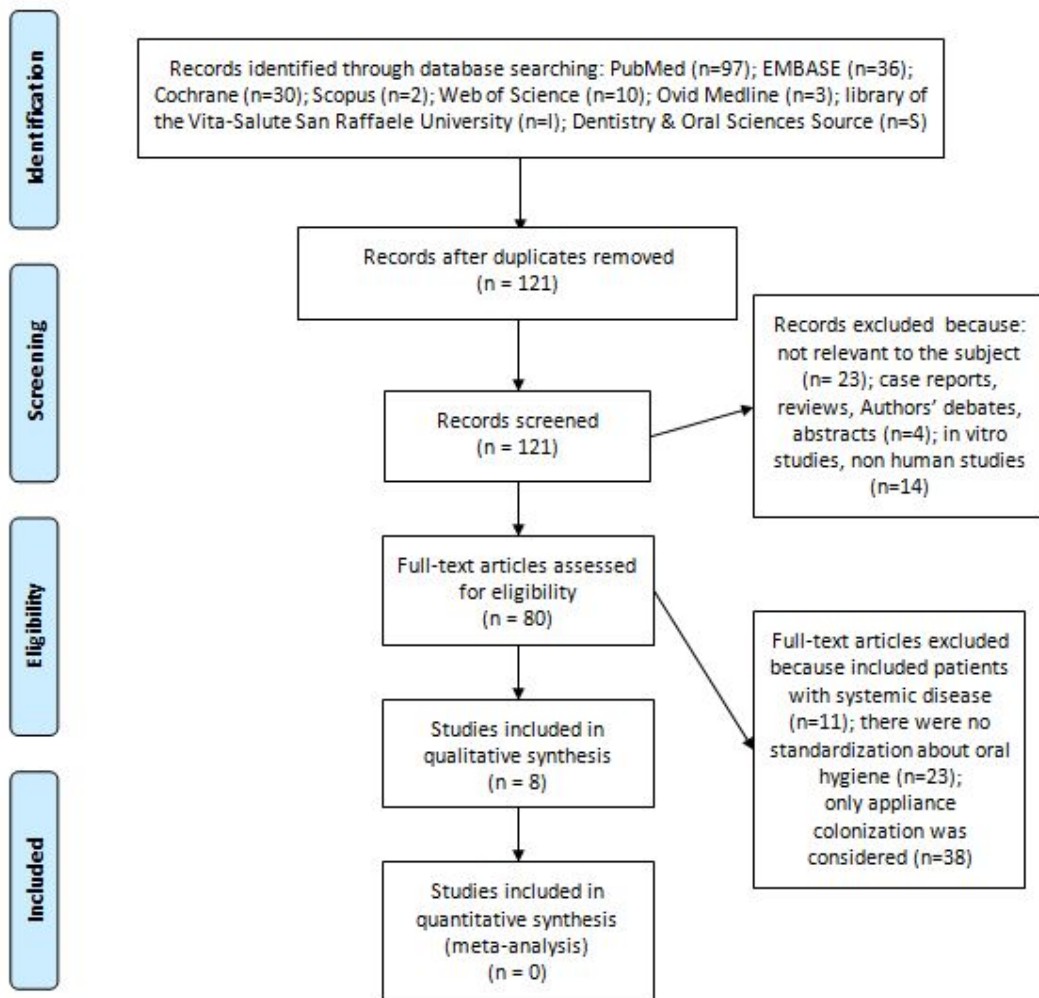

**Figure 1.** PRISMA flow diagram: screening and selection process.

*2.5. Data Selection*

The following data was then collected from each included study: author/year publication, study design, sample size, sample/age/sex, type of appliance, collection time, collection method of analysis, microbial analysis outcome, and quality of the study (Table 1) [11,37–43].

**Table 1.** Characteristics of the included studies.

| Quality of Study (Sbu method) | B | C | B | A |
|---|---|---|---|---|
| **Results** | ↓*S. viridans* ↔*M. catharralis* ↑*S. epidermidis* ↑*Candida* ↑*Lactobacillus* | ↑*S.mutans* ↑*Lactobacillus* | ↑ *Candida* | ↑ *Candida* |
| **Microbial Analisys** | Culture methods | Culture methods | Culture methods | Culture methods |
| **Collection Method** | Sterile swabs | Saliva samples | Square foam-pads | Sterile palstic foam pads |
| **Collection Time** | 2–8 week 2–4 months | Before starting 1 month 3 months 6 months | Before starting 5 months after the insertion 5 months after the removal | Before starting 1 month 3 months |
| **Appliance** | Removable orthodontic appliance | Removable orthodontic appliance | Removable orthodontic appliance | Removable orthodontic appliance |
| **Sample Size/Age/Sex** | 45 p 6–10 y NS | 69 p 6–17 y 31 F 38 M | 33 p 8–27 y 18 F 15 M | 103 p 12–16 y 68 F 35 M |
| **Study Design** | Prospective study | Observational longitudinal study | Longitudinal study | Case-control study |
| **Author/Year** | Jabur, S.F. 2008 [11] | Topaloglu, A. 2011 [43] | Arendorf, T 1985 [40] | Addy, M. 2016 [38] |
| **Quality of Study (Sbu Method)** | B | B | C | B |
| **Results** | ↑*S. mutans* | ↓*Candida* | ↑*Lactobacillus* ↑*Candida* ↑*S.mutans* | ↑*Spirochetes* ↑G+ cocci ↔*Aa* |
| **Microbial Analisys** | Culture methods | Culture methods | Culture methods | Culture methods |
| **Collection Method** | Sterile swabs | Saliva samples | Saliva samples | Sterile swabs Sterile curette (subgingival) |
| **Collection Time** | At least after 6 months | 6 months | Before starting 1 month 3 months 6 months | Before starting 6–8 weeks 6–7 months |
| **Appliance** | Removable orthodontic appliance | Removable orthodontic appliance | Removable orthodontic appliance | Removable orthodontic appliance |

| Sample Size/Age/Sex | 53 p 8–10 y 29 F 24 M | 40 p 11.7 y NS | 20 p 6–15 y NS | 15 p 7–15 y NS |
|---|---|---|---|---|
| **Study Design** | Longitudinal study | Longitudinal study | Longitudinal study | Longitudinal study |
| **Author/ Year** | Batoni, G 2001 [37] | Khanpayeh E. 2014 [39] | Kundu, R. 2016 [41] | Petti, S. 1997 [42] |

↑: increase; ↓: decrease; ↔: no changes; *S. viridans: Streptococcus viridans; M. catharralis: Moraxella catharralis; S. epidermidis: Staphylococcus epidermidis; S. mutans: Streptococcus mutans*; Aa: *Aggregatibacter Actinomycetemcomitans*; NS: not specified; y: years; p: patients; M: male; F: female.

### *2.6. Risk of Bias and Quality Analysis*

The methodological quality is "the extent to which the design and conduct of a study are likely to have prevented systematic errors (bias)." Different quality criteria can explain variation in the results of studies included in a systematic review. More rigorously designed (better "quality") trials are more likely to reach results that are closer to the "truth". The Swedish Council on Technology Assessment in Health Care Criteria for Grading Assessed Studies (SBU) method was adopted to report the level of evidence of this systematic review [44]. To minimize the risk of bias during the inclusion of studies in the analysis, the two reviewers (LA and MM) applied independently the SBU criteria. When there was any disagreement concerning the relevance of an article, it was solved by the intervention of a third reviewer (MMa). This Protocol organized the articles in three grades according to their methodological quality, as Table 2 shows.

**Table 2.** The Swedish Council on Technology Assessment in Health Care Criteria for Grading Assessed Studies (SBU) criteria for grading assessed studies.

| SBU Criteria for Grading Assessed Studies |
|---|
| **Grade A:** high value of evidence. All criteria should be met: randomized clinical study or a prospective study with a well-defined control group, defined diagnosis and endpoints, diagnostic reliability tests and reproducibility tests described, blinded outcome assessment. |
| **Grade B:** moderate value of evidence. All criteria should be met: cohort study or retrospective case series with defined control or reference group, defined diagnosis and endpoints, diagnostic reliability tests and reproducibility tests described. |
| **Grade C:** low value of evidence. One or more of the conditions below: large attrition, unclear diagnosis and endpoints, poorly defined patient material. |

Based on the grade of quality, four evidence levels were used (Table 3).

**Table 3.** Evidence level definition.

| Level | Evidence | Definition |
|---|---|---|
| 1 | Strong | At least two studies assessed at level "A" |
| 2 | Moderate | One study with level "A" and at least two studies at level "B" |
| 3 | Limited | At least two studies at level "B" |
| 4 | Inconclusive | Fewer than two studies at level "B" |

The table shows the criteria used to define the level of evidence of the selected papers.

### 3. Results

From the initial 184 articles, 8 were selected as showed in the PRISMA flow diagram (Figure 1) [11,37–43].

### 3.1. Quality of Evidence

Five of the eight chosen articles presented a moderate methodological quality [11,37,39,40,42]: the major concern regarding these studies is the lack of blinded outcome assessment, diagnostic reliability tests, and reproducibility tests. One article had a high quality [36] and the remaining two were classified as having a low quality [41,43]. Due to the absence of homogeneity in the study formulation, a meta-analysis could not be performed and a systematic review was realized.

When organizing the data according to pathogens, the following results were obtained.

### 3.2. Candida

All studies detected an increase of *Candida* spp. concentration during therapy with removable orthodontic therapy [11,38–41]. According to Jabur et al. study, removable orthodontic appliances induced an increase of *Candida* level up to 13.3% after an average of five weeks and 20% after four months [11].

On the contrary, the increase in *Candida* was very low after three weeks [38] and six months [39].

In Addy's study, the Candida prevalence after three weeks from the beginning of treatment resulted to be 46% in the control group and 52% of removable appliance wearers [38].

Arendorf et al.'s study, noted a prevalence of *Candida* of 57.6% for all study subjects, but the 39.4% of the sample was a prior *Candida*-carrier, so only 18.2% became carriers five months after starting the therapy. Results of McNemar's test showed a highly significant overall increase in *Candida* prevalence while patients were wearing the appliances ($p < 0.001$), especially in posterior and anterior palatal sites, respectively. However, the observed alteration was transient, since removal of the appliance was associated with a highly significant loss of carrier numbers ($p < 0.001$), in fact, five months after the end of the therapy, only 42.4% of subjects reported *Candida* colonies. This observation indicates that removable orthodontic appliances induces a persistent increase of *Candida* colonies of only of 3% [40].

Increasing numbers of microbiological counts of *Candida albicans* were observed from baseline to one, three, and four months after therapy started, with a significant peak at the end of the first month ($p < 0.001$) [41].

According with Khanpayeh et al. study, *Candida* colonies isolated from saliva six months after the beginning of the therapy with removable appliance belonged most frequently to *Candida* spp. (25%) ($p = 0.001$). Colony distribution included: *Candida albicans* 25%, *Candida tropicalis* 3%, *Candida parapsilosis* 2%, *Candida krusei* 1%, and *Candida kefyr* 0%. Though, salivary carrier of *Candida* species decreased with increasing duration of orthodontic treatment [39].

### 3.3. Streptococcus mutans

All three articles [37,41,43] which analyzed *S. mutans* colonization of the mouth agreed that removable orthodontic appliance represents a promoting factor for the colonization of the oral cavity by this microorganism.

In Kundu et al.'s article a statistically significant increase of *S. mutans* was recorded during orthodontic therapy with removable appliances, from the baseline to six months ($p < 0.001$). Furthermore, *S. mutans* bacterial counts were significantly higher than those of *Lactobacillus* spp. and *Candida albicans* at all timepoints (1–3–6 months) [41].

The study that analyzed different interceptive removable appliances [43], demonstrated a constant increase of *Lactobacillus* and an increase of *S. mutans* after 15 days, followed by a progressive decrease after 30 and 60 days.

The numbers of *S. mutans* colonies showed a continuous increase during therapy from baseline to one month with statistical significance ($p < 0.05$) [37].

### 3.4. Lactobacillus

All the studies [11,41,43] which quantitatively and qualitatively evaluated the difference of frequency in *Lactobacillus* spp. demonstrated an increase in the microbiological counts.

Both Kundu et al. and Topaloglu et al. studies suggested that the microscopic counts of *Lactobacillus* spp. increased significantly during orthodontic treatment with removable appliances from baseline to follow-up visits at 1, 3, and 6 months ($p < 0.05$) [41,43].

Jabur et al. noted an increase (6.66%) in *Lactobacillus* spp. after four months of therapy, too [11].

### 3.5. Moraxella catharralis

According to Jabur et al., this pathogen was found in all patients analyzed, furthermore, its oral colonization incredibly increased with removable orthodontic appliances. After a mean of five weeks from the appliance use, *Moraxella* prevalence was of 73.33% and after five months it reached 100% [11].

### 3.6. Staphylococcus epidermidis

*S. epidermidis* colonization of the mouth also appears to be influenced by the use of removable appliances. As Jabur et al. stated, in patients using these devices the percentage increased up to 40% after an average of five weeks and peaked at 60% after four months [11].

### 3.7. Others

The following results revealed that the changes in oral microbiota during treatment with removable orthodontic devices, involved also other bacterial species.

Petti et al. revealed that in supragingival and in subgingival plaque G+ cocci decreased after 6–8 weeks and increased at 6–7 months, with final values higher than baseline values. Gingivitis risk indices (bacterial count and G-rods) significantly increased progressively in 6–8 weeks. Among periodontitis risk indices, only supragingival rods and subgingival Spirochetes significantly increased at 6–7 months. *Aggregatibacter actinomycetemcomitans* (Aa) was nearly absent [40]. Anaerobic bacteria were detected in the subgingival dental plaque with the same density (n = 15.75%) at baseline and at three months, while the prevalence appeared increased, though not reaching statistical significance, at nine months (17.85%). The most important bacteria that cause periodontal tissue loss—*Aggregatibacter actinomycetemcomitans* (Aa), *Porphyromonas gingivalis* (Pg), *Tannerella forsythia* (Tf), and *Prevotella nigrescens*—were not detected in any patients [42].

### 3.8. Outcomes Summary

Removable orthodontic appliances have the following effects on oral microflora:

*Candida* colonies increase, especially *C. albicans* species, during the first month of therapy, followed by a decrease after a few months.

*S. mutans* is the main microorganism to increase during the first months of therapy and the main increment occurs in the first 15 days.

*Lactobacillus* spp. microbiological count increases during the first months of therapy.

*Moraxella catharralis* and *S. epidermidis* values increase significantly during the first month of therapy.

*Spirochaetes* spp. significantly increases during the first 6–7 months of treatment.

Aa, Pg, Tf, and *Prevotella nigrescens* were not detected during therapy.

## 4. Discussion

The present systematic review is based on a low number of selected articles (n = 8) because of the reduced number of papers focused on this topic. Since we favored a strict and accurate study-selection process, the lack of standardization between the studies, the

disparities in the category of devices analyzed and the variability in wearing-time did not allow to carry out a meta-analysis.

Any appliance or device placed in the oral cavity creates new retentive surfaces, promoting plaque accumulation and alteration of oral microflora. Consequently, the pH values and the buffering capacity of saliva significantly reduces during the therapy. This condition can promote an accumulation of cariogenic bacteria in dental plaque and saliva [37]. Statistically significant increases were recorded in the following bacteria: *Streptococcus mutans,* [37,41,43], *Lactobacillus* spp. [11,41,43], *Staphylococcus epidermidis* [11], *Moraxella catharralis* [11], and subgingival Spirochetes [42]. It was interesting to notice that the initial microorganism increment was followed by a progressive decrease towards more physiological values [39]. The same happened to pH values, which seem to return to physiological levels after 6 months from the end of therapy [40].

It was also demonstrated that previous orthodontic therapies do not alter the response of the oral microbiota to removable orthodontic treatment. Indeed, patients who have already undergone orthodontic treatment [29,30] have the same alterations in microorganisms of those who never wore orthodontic appliances [25].

These results emphasize that removable orthodontic appliances, when inserted in the oral cavity, begin to accumulate plaque. However, it is not possible to understand whether the accumulation of plaque may depend on the material from which the device is made, because most published studies do not specify it. The microorganisms load increment could be strictly related to the appliance surface roughness as well as the time spent in the oral cavity. Generally, removable orthodontic appliances are made of heat-setting plastic or acrylic resin, which are both microporous and rough materials. The introduction of smoother surface removable devices could be more resilient for microorganisms and more biocompatible.

Therefore, it is essential that both patients and healthcare professionals embrace the concept of a balanced oral microbiota and its importance to oral and systemic health. Treatment sessions should include prevention strategies that promote active maintenance of oral health, rather than disease management. Oral health professionals can achieve this educating patients to appropriate lifestyle choice and an effective biofilm-formation control. This approach would maintain the beneficial properties of resident microbiota and would reduce the risk of dental disease and fungal infections.

There is a lack of studies rigorously designed to examine changes in the oral microbiota associated with removable orthodontic therapy; furthermore, a lot of heterogeneity was observed in the identified studies; therefore, further research is needed on this topic [45].

## 5. Conclusions

With the limitations related to the studies analyzed and the culture method used, according to our systematic review, removable orthodontic appliances influence some oral bacterial species qualitatively and quantitatively. Moreover, the main changes seem to occur during the first 15 days, independently from the type of appliance. Nevertheless, after the end of the treatment, the concentration of pathogenic microorganisms seems to be leaning towards more physiological values.

Though, these changes promoted by orthodontic removable appliances appear to be transient.

**Author Contributions:** Conceptualization, A.L. and M.Ma Maurizio Manuelli; methodology, E.G.; software, M.T.L.S. and A.B.; validation, C.B., M.T.L.S., and F.C.; formal analysis, A.L. and P.A.; investigation, A.L.; resources, M.Ma Maurizio Manuelli; data curation, M.N. and F.C.; writing—original draft preparation, M.M. Marta Marcolina and E.T.; writing—review and editing, V.B. and J.P.; visualization, E.G.; supervision, C.B. and R.G.; project administration, M.Ma Maurizio Manuelli. All authors have read and agreed to the published version of the manuscript.

**Funding:** This research received no external funding.

**Institutional Review Board Statement:** The study was conducted according to the guidelines of the Declaration of Helsinki, and approved by the Ethics Committee of IRCCS San Raffaele Scientific Institute, Milan, Italy (107/1NT/2017).

**Informed Consent Statement:** Not applicable.

**Data Availability Statement:** No new data were created or analyzed in this study. Data sharing is not applicable to this article.

**Conflicts of Interest:** The authors declare no conflict of interest.

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
