# Peer review of "The Effect of Removable Orthodontic Appliances on Oral Microbiota: A Systematic Review"

_applsci, doi:10.3390/app11062881_

Round 1

Reviewer 1 Report

The paper itself is well written and documented, showing a great effort from the authors.

I would make only the following few mentions:

Abstract: Is strongly encouraged to use a structured abstract with numbers between brackets indicating background, methods, results, conclusions.

Line 81: Library of the Vita salute San Raffaele is mentioned in the PRISMA flow chart but not in the methods section. Furthermore, Gray literature (for instance Open Gray) was not searched.

Line 83: Please add the PICOS strategy you have used. You can add it as Supplementary materials.

Line 170: Please add a reference for this tool: “Swedish Council on Technology 170 Assessment in Health Care Criteria for Grading Assessed Studies’(SBU)”

Table 1: Please add the references to the articles described in the table.

References:

  • N° 26 Use abbreviated journal name.

It is suggested a revision of the English used in the manuscript.

Otherwise, the article is good.

Author Response

Reviewer 1: The paper itself is well written and documented, showing a great effort from the authors.

I would make only the following few mentions:

Abstract: Is strongly encouraged to use a structured abstract with numbers between brackets indicating background, methods, results, conclusions.

Ok we used a structured abstract with numbers between brackets.

Line 81: Library of the Vita salute San Raffaele is mentioned in the PRISMA flow chart but not in the methods section. Furthermore, Gray literature (for instance Open Gray) was not searched.

We correct the manuscript with the databases mentioned.

Line 83: Please add the PICOS strategy you have used. You can add it as Supplementary materials.

We add the PICOS strategy used.

Line 170: Please add a reference for this tool: “Swedish Council on Technology 170 Assessment in Health Care Criteria for Grading Assessed Studies’(SBU)”

Ok, we add the new reference

Table 1: Please add the references to the articles described in the table.

Ok added the references

References:

  • N° 26 Use abbreviated journal name.

Ok we change it with the abbreviated name.

It is suggested a revision of the English used in the manuscript.

Yes we did

Otherwise, the article is good.

Response 1: We made all the corrections needed.

Reviewer 2 Report

This systematic review, although simplified and lacking meta-analysis, seems to be well executed and correctly planned. 

Major comments:

  1. Better description of inclusion criteria required
  2. Verification of heterogenecity found 
  3. More comprehensive discussion regarding a potential bias  

Minor comments: 

  1. Microorganisms could be combined considering their common taxonomic origin   
  2. Typos need corrections

Author Response

Reviewer 2: This systematic review, although simplified and lacking meta-analysis, seems to be well executed and correctly planned. 

Major comments:

  1. Better description of inclusion criteria required
  2. Verification of heterogenecity found 
  3. More comprehensive discussion regarding a potential bias  

Minor comments: 

  1. Microorganisms could be combined considering their common taxonomic origin   
  2. Typos need corrections

Ok we made the corrections suggested.

Response 2: We thank the reviewer for his comments; we implemented inclusion criteria and highlighted the lack of heterogeneity in the identified studies, all possible biases related to the study were highlighted and grammatical corrections were made with a native speaker.

Reviewer 3 Report

Unfortunately, the manuscript is poor. Many sentences do not connect with each other and the article is written laboriously. In review are included only eight publications. In all of these microbial analysis was made using culture method. Authors, in 4 publications presented change in one microbial species, in next 4 publications change from 2 to 4 species. Microbiota is ecological community of commensal, symbiotic and pathogenic microorganisms. It is difficult to name 1 or 2 species as a microbiota. One species is not community. In review is lack of publications in which were used other methods of microbial detection, e.g. NGS for microbiome detection. In culture method we detect only some microbial species, in NGS we can detect several dozen microorganisms, including anaerobic and non-cultivable. Many articles showed that in oral appliances rise the levels of Enterobacterales and Enterococcus, but in this review are not data about it.

Author Response

Reviewer 3: Unfortunately, the manuscript is poor. Many sentences do not connect with each other and the article is written laboriously. In review are included only eight publications. In all of these microbial analysis was made using culture method. Authors, in 4 publications presented change in one microbial species, in next 4 publications change from 2 to 4 species. Microbiota is ecological community of commensal, symbiotic and pathogenic microorganisms. It is difficult to name 1 or 2 species as a microbiota. One species is not community. In review is lack of publications in which were used other methods of microbial detection, e.g. NGS for microbiome detection. In culture method we detect only some microbial species, in NGS we can detect several dozen microorganisms, including anaerobic and non-cultivable. Many articles showed that in oral appliances rise the levels of Enterobacterales and Enterococcus, but in this review are not data about it.

we thank the reviewer for his comment; the articles that have been included in the review are not numerous because unfortunately, given the strict criteria of inclusion and exclusion, many studies did not fit into our analysis. In the literature, the orthodontic appliances that are analyzed in the various studies are very heterogeneous one from each other; however, if we want to focus only on removable devices, the articles with a good scientific methodology belonging to this category are not numerous and are mainly carried out in culture.

Reviewer 4 Report

Nice and interesting study!

Author Response

Thanks for your comments.

Round 2

Reviewer 3 Report

Thank you for Authors' response. I understand that in the review are only some articles included. But, unfortunately, Authors not corrected the manuscript, which is written laboriously. Moreover, Authors not presented microbiota, because most of cited articles described 1 or 2 species. One or 2 species are not microbiota. In review is lack of publications in which were used sequencing, presented real microbiota.

Due to poor descriptions, misunderstanding of what microbiota is, and submitting articles with 1 or 2 species instead of the actual microbiota, I suggest that the authors first thoroughly correct the manuscript, and then send it back to the editor.